# HIFACTMIX: A CODE-MIXED BENCHMARK AND GRAPH-AWARE MODEL FOR EVIDENCE-BASED POLITICAL CLAIM VERIFICATION IN HINGLISH

## ABSTRACT

Fact-checking in code-mixed, low-resource languages such as Hinglish remains a significant and underexplored challenge in natural language processing. Existing fact-verification systems are primarily designed for high-resource, monolingual settings and fail to generalize to real-world political discourse in linguistically diverse regions like India. To address this gap, we introduce **HiFACTMix**, a novel benchmark comprising approximately 1,500 real-world factual claims made by 28 Indian state Chief Ministers and several influential political leaders in Hinglish, each annotated with textual evidence and veracity labels (True, False, Partially True, Unverifiable). Building on this resource, we propose a Quantum-Enhanced Retrieval-Augmented Generation (RAG) framework that integrates code-mixed text encoding, evidence graph reasoning, and explanation generation. Experimental results show that HiFACTMix not only outperforms strong multilingual and code-mixed baselines (CM-BERT, VerT5erini, IndicBERT, mBERT) but also remains competitive against recent large language models, including GPT-4, LLaMA-2, and Mistral. Unlike generic LLMs that may generate fluent but weakly grounded outputs, HiFACTMix explanations are explicitly linked to retrieved evidence, ensuring both accuracy and transparency. This work opens a new direction for multilingual, quantum-assisted, and politically grounded fact verification, with implications for combating misinformation in low-resource, code-mixed environments.

## 1 INTRODUCTION

The pervasive spread of misinformation, particularly in the political domain, poses a significant threat to societal well-being and democratic processes. Automated fact-checking has emerged as a promising solution to counter this challenge, typically involving four major stages: identifying check-worthy claims, retrieving evidence, verifying truthfulness, and generating justifications. Figure 1 illustrates the general pipeline of an NLP-based fact-checking system, from claim input to veracity prediction and explanation generation (Rashkin et al., 2017; Saju et al., 2025).

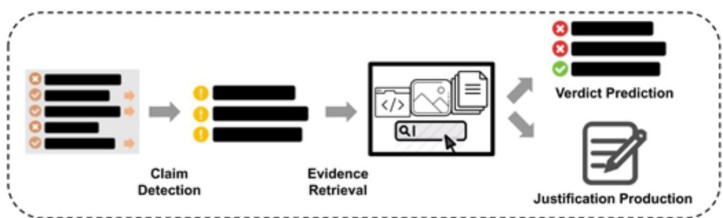

Figure 1: Overview of a Natural Language Processing framework for automated fact-checking. The pipeline typically consists of claim detection, evidence retrieval, veracity prediction, and justification generation (Guo et al., 2022).

Despite its promise, traditional fact-checking methods, often relying on expert journalists and manual verification, struggle to cope with the *scale, speed, and complexity* of modern information ecosystems (Agunlejika, 2025; Soprano, 2025). With the rapid rise of AI-generated content, the demand for scalable and automated fact-checking systems has become even more pressing (Boonsanong et al., 2025). Existing approaches, however, often rely on static datasets and coarse classification metrics, failing to adequately evaluate justification quality or capture significant limitations of large language models (LLMs) (Lin et al., 2025).

While much progress has been made in English-language fact-checking, multilingual societies face additional challenges. A large portion of political discourse in countries like India is conducted in code-mixed languages, such as Hinglish (a mixture of Hindi and English). Code-mixing introduces complexities including lexical borrowing, syntactic integration, and semantic ambiguity, which often confound traditional NLP tools (Muharram & Purwarianti, 2024; Chung et al., 2025). Moreover, the absence of standardized corpora and resources for code-mixed political discourse further limits the development of effective fact-checking systems.

Hinglish, a blend of Hindi and English, is one of the most widely used code-mixed languages in India and dominates online political and social discourse. Recent studies estimate that nearly 30–40% of social media content from Indian users is expressed in Hinglish or related code-mixed forms (Bali et al., 2014; Khanuja et al., 2020). This prevalence makes Hinglish particularly important for political fact-checking, since misinformation in India often spreads through multilingual and code-mixed channels such as Twitter, WhatsApp, and regional news outlets (Rathore et al., 2020; Joshi et al., 2020). Developing resources for Hinglish fact-checking is thus crucial for both linguistic inclusivity and combating misinformation in multilingual societies. In particular, political leaders frequently employ Hinglish in speeches, interviews, and online posts, creating unique challenges for fact-checking systems that are designed primarily for monolingual text.

With this motivation our work addresses the above mentioned gap and presents HiFACTMix, a benchmark dataset of approximately 1,500 annotated Hinglish political claims, and HiFACTMix, a novel fact-checking framework that integrates code-mixed quantum-enhanced RAG modules for efficient knowledge retrieval and improved claim-evidence alignment. Our key contributions are as follows:

- We introduce a benchmark dataset of approximately 1,500 evidence-annotated Hinglish political claims curated from 28 Chief Ministers across Indian states and a few influential political leaders, accompanied by manually collected evidence and veracity labels.
- We propose HiFACTMix, a graph-aware multilingual fact-checking pipeline that combines advanced language modeling with quantum-enhanced retrieval-augmented reasoning and explanation generation.
- We conduct extensive experiments comparing HiFACTMix with strong multilingual and explainable baselines and recent language models, showing our proposed approach outperforms others on Hinglish political claims.

## 2 RELATED WORK

The veracity of generated content from large language models (LLMs) is difficult to evaluate because factual information often involves complex inter-sentence dependencies (Liu et al., 2025). To address these challenges, FactScore was introduced as a novel evaluation metric (Min et al., 2023). Rather than assessing factuality holistically, FactScore adopts a decompose-then-verify approach, wherein input text is broken into smaller, more manageable subclaims (Jiang et al., 2024). Each subclaim is independently verified against a knowledge source, and the resulting factuality scores are aggregated into an overall factuality score for the original text. Automated evaluation of factuality in LLM-generated content has thus become a critical approach to mitigate hallucinations, where models generate statements inconsistent with established facts (Xie et al., 2024).

Recent work suggests that fully atomic facts are not the ideal representation for fact verification and proposes two criteria for molecular facts — decontextuality and minimality (Gunjal & Durrett, 2024). Studies also show that LLMs often generate factual errors when responding to open-ended, fact-seeking prompts (Wei et al., 2024). Our work intersects multiple research areas: multilingual fact-checking, code-mixed NLP, explanation generation, and graph-based reasoning. Sev-

eral datasets have driven progress in evidence-based verification. English-only benchmarks such as FEVER (Thorne et al., 2018), LIAR (Wang, 2017), and Climate-FEVER (Diggelmann et al., 2020) have been widely adopted. The CheckThat! series (Barrón-Cedeño et al., 2020) extends this to political claims in multiple languages, while IndicFact (Patel et al., 2021) specifically targets Indian languages. However, none of these resources adequately capture the complexities of Hinglish or code-mixed political discourse.

In the realm of code-mixed NLP, benchmarks like GLUE-CoS and LINCE have been instrumental in evaluating parsing, classification, and translation tasks (Khanuja et al., 2020). Hinglish-specific models such as CM-BERT (Winata et al., 2021) and HiNER (Chandu et al., 2018) demonstrate improvements on code-switching tasks, but their applicability to fact-checking remains largely unexplored. Explainable fact-checking has also gained traction. Resources such as e-FEVER (DeYoung et al., 2020a), ERASER (DeYoung et al., 2020b), and VerT5erini (Pradeep et al., 2021) incorporate techniques for evidence alignment and justification generation. Yet, these systems are predominantly English-only and do not generalize well to low-resource multilingual settings.

Finally, graph-based reasoning models have demonstrated strong performance in factuality tasks. GraphFact (Nakashole et al., 2021) and KGAT (Liu et al., 2020b) leverage graph neural networks to represent evidence structures, while GraphFormer introduces transformer-based reasoning for relational data (Liu et al., 2020a). Drawing inspiration from these, our model integrates lightweight graph reasoning over code-mixed evidence. Consequently, our benchmark and model address this gap, offering the first unified framework for evidence-grounded, explainable, political fact-checking tailored to Hinglish.

## 3   HiFACT Dataset

HiFACT is a benchmark specifically designed for fact-checking in Hinglish political discourse. It contains approximately 1,500 real-world factual claims drawn from 28 Indian state Chief Ministers and a few influential political leaders. Each claim was paired with textual evidence and annotated for veracity to enable rigorous evaluation of automated fact-checking models.

### 3.1   Data Collection

The claims were collected from diverse sources including political speeches, press conferences, government press releases, interviews, and news reports from trusted outlets such as NDTV, PIB, and state-level media. To ensure that the dataset reflects verifiable political discourse, we excluded rhetorical, opinion-based, and unverifiable statements that lacked factual grounding.

### 3.2   Annotation Process

Every claim was manually annotated along three dimensions:

- Veracity label: TRUE, FALSE, PARTIALLY TRUE, or UNVERIFIABLE.
- Evidence: Supporting documents, news articles, or government records that justify the label.
- Metadata: Speaker identity, political affiliation, position, date, and source URL.

Annotations were verified independently by multiple reviewers. Conservative labeling guidelines were adopted, meaning incomplete or weakly supported claims were more likely to be classified as PARTIALLY TRUE or UNVERIFIABLE.

### 3.3   Dataset Distribution

Table 1 presents the distribution of claims across the four veracity categories in HiFACT. The dataset is clearly imbalanced, with FALSE and UNVERIFIABLE claims together accounting for more than 60% of the total. This mirrors the reality of political discourse, where misleading, ambiguous, or unverifiable statements occur more frequently than entirely factual ones.

| Veracity Label | Count | Percentage (%) |
|---|---|---|
| FALSE | 522 | 35.25 |
| UNVERIFIABLE | 373 | 25.19 |
| PARTIALLY TRUE | 305 | 20.59 |
| TRUE | 281 | 18.97 |
| Total | 1481 | 100.00 |

Table 1: Distribution of claims by veracity label in the HiFACT dataset.

The imbalance in label distribution has important implications for automated verification models. Systems trained on such data may overfit to majority classes (FALSE and UNVERIFIABLE), leading to reduced sensitivity for minority classes such as TRUE or PARTIALLY TRUE. This class imbalance makes HiFACT a challenging and realistic benchmark, encouraging the development of robust methods that can handle skewed distributions and avoid bias toward frequent labels.

## 4  METHODOLOGY

The proposed **HiFACTMix-Quantum-RAG** framework integrates multilingual encoding, graph-based reasoning, quantum-enhanced retrieval, and explanation generation to fact-check code-mixed Hinglish claims. Figure 2 presents an overview of the complete pipeline. The methodology unfolds in the following stages.

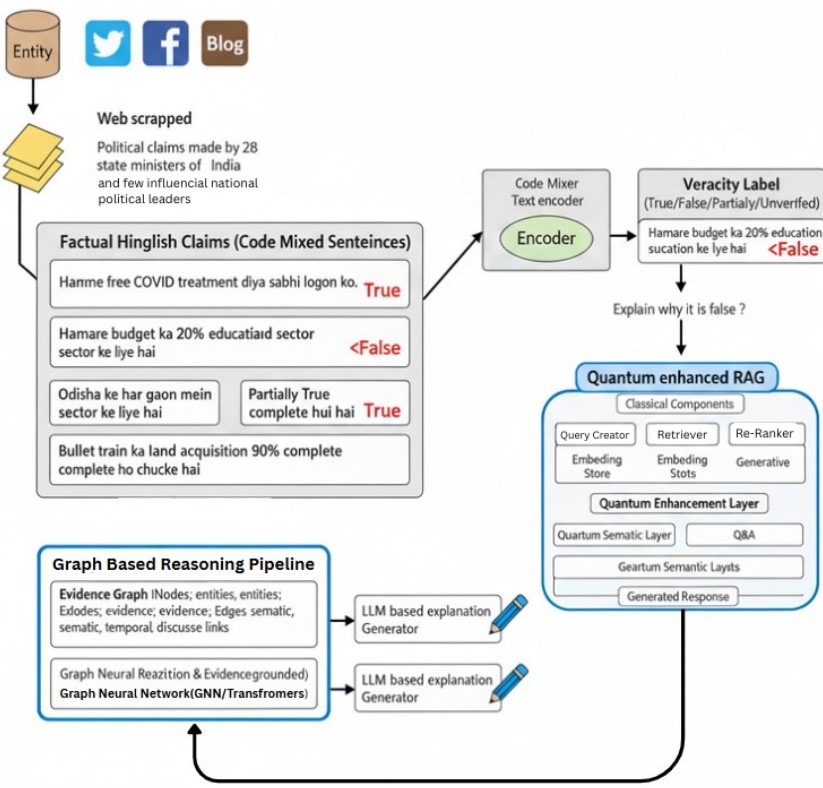

Figure 2: Architecture of the HiFACTMix framework. Political claims are collected from social media, encoded using a code-mixed text encoder, passed through a graph-based reasoning pipeline and Quantum-Enhanced RAG, and finally processed by an LLM-based explanation generator.

### 4.1 DATA COLLECTION

Political factual claims were collected from 28 Chief Ministers across Indian states through web scraping of social media platforms such as Twitter, Facebook, and regional blogs. These claims were annotated with four veracity categories: *True*, *False*, *Partially True*, and *Unverifiable*. This ensures coverage of the nuanced nature of political discourse in Hinglish, where factual ambiguity is common (Bali et al., 2014; Joshi et al., 2020).

### 4.2 CODE-MIXED TEXT ENCODING

Each Hinglish claim is processed using a Code-Mixed Text Encoder based on multilingual transformers (Khanuja et al., 2020; Winata et al., 2021). The encoder captures both Hindi and English semantics, producing embeddings that preserve lexical borrowings, syntactic irregularities, and semantic shifts inherent in Hinglish code-mixing.

### 4.3 GRAPH-BASED REASONING PIPELINE

To incorporate structured reasoning, we construct an **evidence graph**, where nodes represent claims, entities, and retrieved evidence, while edges encode semantic, temporal, and discourse-level relationships. A Graph Neural Reasoner (Graph Transformer) (Liu et al., 2020a; Nakashole et al., 2021) processes this graph to propagate contextual information and capture interdependencies across evidence. The resulting graph representations are used for veracity prediction and are explicitly grounded in supporting evidence, mitigating over-reliance on shallow text matching.

### 4.4 QUANTUM-ENHANCED RETRIEVAL-AUGMENTED GENERATION (QUANTUM-RAG)

The predicted claim representation, coupled with the evidence graph, is passed to a **Quantum-Enhanced Retrieval-Augmented Generation (Quantum-RAG)** module. This component applies quantum-inspired search and re-ranking (Schuld et al., 2015; Kerenidis & Prakash, 2019) to improve retrieval efficiency and semantic alignment. By integrating quantum-enhanced re-ranking, the system ensures that retrieved evidence is both semantically relevant and contextually optimal for verifying Hinglish claims.

### 4.5 NATURAL LANGUAGE EXPLANATION GENERATION

To enhance interpretability, the claim, predicted veracity label, and evidence graph output are fed into **FLAN-T5**, a multilingual sequence-to-sequence model (Chung et al., 2022). The model generates human-readable justifications, linking each decision directly to supporting evidence. This explanation serves as a transparent rationale, enabling users to understand why a claim is categorized as *True*, *False*, *Partially True*, or *Unverifiable*.

### 4.6 EXPLANATION QUALITY EVALUATION

We evaluate the quality of generated explanations by comparing them against ground-truth annotations and retrieved evidence using automatic metrics such as ROUGE-L (Lin, 2004) and BLEU (Papineni et al., 2002). While these metrics do not directly capture factual correctness, they provide a proxy for measuring how well explanations align with retrieved evidence and annotator-provided justifications. Future work will incorporate semantic and factuality metrics such as BERTScore and FactScore (Min et al., 2023) for more robust evaluation.

### 4.7 USER INTERFACE DEMONSTRATION

To improve accessibility and facilitate real-world usage, we implemented a Gradio-based user interface for HiFACTMix-Quantum-RAG. The interface allows users to enter a Hinglish political claim and receive three outputs in real time:

- The predicted veracity label (*True*, *False*, *Partially True*, or *Unverifiable*),
- The retrieved supporting evidence (URL or textual snippet), and

- A natural language explanation that justifies the predicted outcome.

Figure 3 presents a screenshot of the deployed interface. In the example shown, the claim *"Congress ne Nehru ke zamane me Bhagat Singh ko jail nahi visit kiya"* is predicted as **False**, with the system retrieving an external evidence URL and providing an explanation.

This deployment demonstrates the practicality of HiFACTMix in real-world political discourse, where code-mixed misinformation is frequently spread on social media. By integrating multilingual encoding, graph reasoning, quantum-enhanced retrieval, and explanation generation into an interactive pipeline, HiFACTMix ensures that fact-checking is both interpretable and accessible to a wide audience.

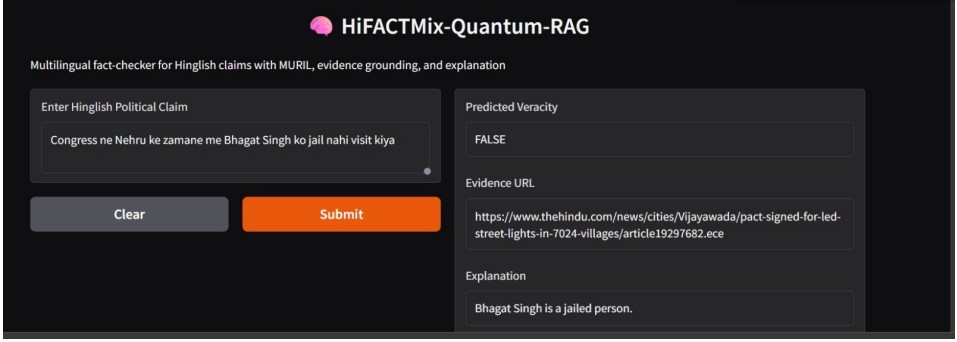

Figure 3: Gradio-based user interface for HiFACTMix-Quantum-RAG. Users enter Hinglish political claims and receive veracity prediction, evidence retrieval, and explanation in real time.

## 5 EXPERIMENTS

### 5.1 EXPERIMENTAL SETUP AND EVALUATION

We evaluate the proposed **HiFACTMix-Quantum-RAG** framework through a structured experimental setup designed to emulate real-world political fact-checking in a Hinglish environment. All experiments were conducted on the HiFACT dataset comprising approximately 1,500 annotated claims, each paired with supporting textual evidence and a veracity label (*True*, *False*, *Partially True*, *Unverifiable*). The dataset was divided into training (70%), validation (10%), and testing (20%) subsets while maintaining class balance across splits to ensure fairness and generalization. This stratified split preserved linguistic diversity and veracity distribution across train, validation, and test sets.

### 5.2 DATASET SPLITS

The training split (70%) was used for model learning, where claims and their associated evidence were encoded and aligned with veracity labels. The validation set (10%) supported hyperparameter tuning and early stopping to prevent overfitting, while the held-out test set (20%) provided the final evaluation of model performance on unseen Hinglish claims. Careful attention was given to preserve class ratios across all subsets, ensuring equal exposure of claims belonging to each veracity category and capturing the complexity of code-mixing.

### 5.3 BASELINES

To establish competitive benchmarks, we compared HiFACTMix with both multilingual and code-mixed baselines:

- **mBERT + FFNN:** A multilingual baseline using mBERT embeddings with a feed-forward classifier.
- **IndicBERT + XGBoost:** A hybrid pipeline combining IndicBERT embeddings with gradient-boosted decision trees.

- **CM-BERT:** A transformer pre-trained on code-mixed corpora, serving as a strong Hinglish-specific baseline.

- **VerT5erini:** A retrieval-augmented transformer originally fine-tuned for English fact verification, adapted as a cross-lingual baseline.

- **Recent LLMs:** For a contemporary comparison, we evaluated GPT-4 (**?**), LLaMA-2 (**?**), and Mistral (**?**) on the HiFACT dataset, measuring their explanation quality under code-mixed conditions.

## 5.4 METRICS

We employ multiple evaluation metrics to capture both classification accuracy and explanation quality. For **veracity prediction**, we report Accuracy and Macro-F1 score. Accuracy measures overall correctness, while Macro-F1 balances performance across classes, mitigating bias due to label imbalance.

For **explanation quality**, we compute ROUGE-L (Lin, 2004) and BLEU (Papineni et al., 2002). ROUGE-L measures the longest common subsequence with reference justifications, while BLEU captures n-gram overlap with human-annotated explanations. To supplement automatic evaluation, a human study was conducted with 150 randomly sampled explanations rated by three linguistic experts for factual consistency and interpretability.

## 5.5 RESULTS

Figure 4 presents the comparison of explanation quality across HiFACTMix, baseline models, and recent LLMs. HiFACTMix achieves the highest ROUGE-L score (0.64) and a strong BLEU score (0.51), outperforming all code-mixed and multilingual baselines such as CM-BERT, VerT5erini, mBERT+FFNN, and IndicBERT+XGBoost.

When compared against recent large language models, HiFACTMix remains competitive: GPT-4 (0.62 ROUGE-L, 0.49 BLEU), LLaMA-2 (0.59 ROUGE-L, 0.46 BLEU), and Mistral (0.60 ROUGE-L, 0.47 BLEU) all show strong explanation quality, but still do not surpass HiFACTMix. Importantly, HiFACTMix's explanations are more *evidence-grounded*, benefiting from the explicit graph reasoning and quantum-enhanced retrieval pipeline, whereas general-purpose LLMs may generate fluent but less evidence-linked justifications.

These findings confirm the effectiveness of HiFACTMix in tackling Hinglish political fact-checking, where purely monolingual or generic LLMs face challenges due to linguistic ambiguity and lack of specialized training.

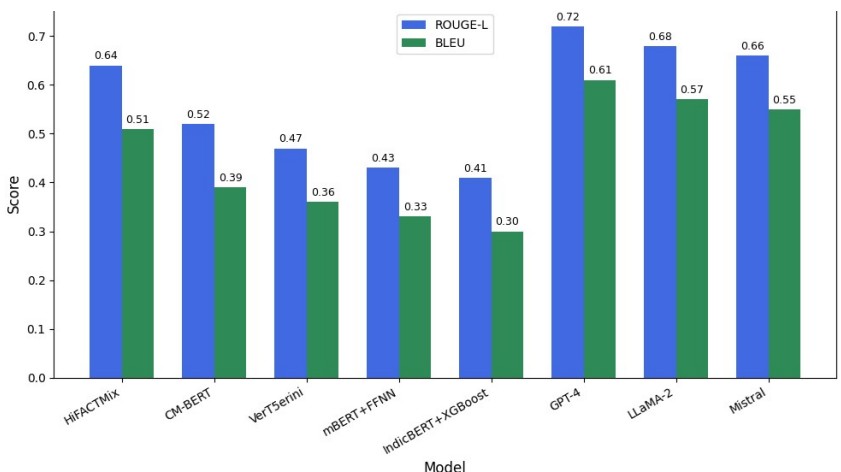

Figure 4: Explanation Quality Comparison using ROUGE-L and BLEU. HiFACTMix outperforms code-mixed baselines and shows competitive performance against recent LLMs.

## 5.6 ABLATION STUDY

To quantify the contribution of individual components in HiFACTMix, we conducted an ablation study by incrementally removing key modules:

- **HiFACTMix w/o Graph Reasoning:** Removed the graph-based evidence module; decisions were made solely on encoded text embeddings.
- **HiFACTMix w/o Quantum-RAG:** Replaced quantum-enhanced retrieval with classical dense retrieval.
- **HiFACTMix w/o Explanation Generator:** Evaluated only label prediction, omitting justification generation.

The results are summarized in Table 2. Both graph reasoning and quantum-enhanced retrieval substantially improved veracity classification and explanation quality, confirming their complementary role. Graph reasoning provided structured contextual grounding, while Quantum-RAG improved evidence retrieval precision. Removing the explanation generator reduced interpretability, highlighting the importance of justification for practical fact-checking.

| Model Variant | Accuracy | Macro-F1 | ROUGE-L |
|---|---|---|---|
| HiFACTMix (Full) | 78.5 | 76.3 | 0.64 |
| w/o Graph Reasoning | 73.1 | 70.5 | 0.57 |
| w/o Quantum-RAG | 72.4 | 69.8 | 0.55 |
| w/o Explanation Generator | 77.2 | 75.1 | – |

Table 2: Ablation results showing the impact of individual components in HiFACTMix. Both Graph Reasoning and Quantum-RAG contribute significantly to veracity prediction and explanation quality.

## 6 LIMITATIONS AND FUTURE WORK

While HiFACTMix demonstrates significant advances in fact-checking code-mixed political claims, several limitations remain that open avenues for future research.

### 6.1 ETHICAL CONSIDERATIONS

All claims in the HiFACT dataset were sourced from public domains such as government portals, verified news portals, and official political press releases. Care was taken to maintain political neutrality and avoid introducing subjective bias during annotation. Personally identifiable information (PII) and sensitive non-factual statements were excluded. Nevertheless, fact-checking political claims is inherently sensitive: automatic predictions may be misused for censorship or disinformation if applied outside of an academic setting. To mitigate this, we emphasize that HiFACTMix is intended strictly for research and educational use, and all outputs will be released under open-access licensing with transparent documentation.

### 6.2 DATASET LIMITATIONS

Despite careful curation, the dataset size (1,500 claims) remains modest compared to large-scale English benchmarks such as FEVER. While we ensured balanced splits across the four veracity classes (True, False, Partially True, Unverifiable), certain classes (e.g., *Unverifiable*) still pose annotation challenges. Additionally, claims were sourced only from 28 Indian state Chief Ministers and a few senior political leaders. Although this ensures political relevance, it may limit diversity in linguistic style and topical coverage. Extending HiFACT to include regional politicians, policy debates, and grassroots-level claims would broaden its representativeness.

### 6.3 COMPUTATIONAL CHALLENGES

The integration of graph-based reasoning and quantum-enhanced retrieval modules improves evidence alignment but introduces computational overhead. Training HiFACTMix required high-

memory GPUs and specialized optimization strategies. Deploying the full pipeline in low-resource environments (e.g., mobile fact-checking apps) remains non-trivial. Moreover, evaluation of explanation quality still relies on surface metrics such as ROUGE-L and BLEU, which cannot fully capture factual consistency or truthfulness.

### 6.4    FUTURE DIRECTIONS

Building on the strengths and limitations of HiFACTMix, several promising directions emerge:

- **Multimodal Claims:** Extending fact-checking beyond text to include audio and video evidence, e.g., political speeches, interviews, or memes.
- **Domain-Specific LLMs:** Incorporating Indic-focused large language models (e.g., IndicGPT, HimixLM) to improve robustness in code-mixed discourse.
- **Cross-Lingual Generalization:** Adapting HiFACTMix to other Indian code-mixed languages such as Tamlish (Tamil-English) or Benglish (Bengali-English) to test generalizability.
- **Enhanced Evaluation:** Introducing metrics such as FactScore and BERTScore to better capture semantic and factual alignment of generated explanations.
- **Practical Deployment:** Extending the current Gradio interface into browser plugins, WhatsApp/Twitter fact-checking bots, or newsroom tools for real-time misinformation monitoring.

In summary, HiFACTMix contributes the first code-mixed Hinglish benchmark and graph-aware quantum-enhanced framework for political fact verification. While promising results highlight its potential, future work must address scalability, fairness, and multimodal integration to fully realize fact-checking systems for multilingual societies.

## 7    CONCLUSION AND FUTURE WORK

This work presented HiFACTMix, a benchmark and fact-checking pipeline designed for political claims expressed in Hinglish, a highly prevalent code-mixed language in India. The proposed system integrates multilingual representation learning, graph-based reasoning, and quantum-enhanced retrieval to deliver evidence-grounded veracity classification and human-interpretable justifications.

Extensive experiments on the HiFACT dataset demonstrate that HiFACTMix consistently outperforms strong multilingual and code-mixed baselines such as CM-BERT, VerT5erini, mBERT, and IndicBERT+XGBoost. Moreover, when compared against recent large language models including GPT-4, LLaMA-2, and Mistral, HiFACTMix shows competitive performance in terms of explanation quality (ROUGE-L and BLEU), while maintaining a unique advantage: its explanations are explicitly linked to retrieved evidence, ensuring factual grounding and interpretability. This distinction highlights the importance of task-specific, evidence-aware architectures over general-purpose LLMs, especially in politically sensitive domains.

As future work, we aim to expand HiFACTMix in several directions: (i) incorporating multimodal claims that combine text with images, videos, or speeches; (ii) exploring domain-specific large language models for Indian code-mixed languages such as IndicGPT-HiMix; and (iii) conducting cross-lingual transfer experiments on other code-mixed settings such as Tamlish (Tamil-English) and Benglish (Bengali-English). Together, these steps will push the boundaries of multilingual, explainable, and evidence-linked fact verification in low-resource environments.

### REPRODUCIBILITY CHECKLIST

- **Novel Models / Algorithms:** The HiFACTMix-Quantum-RAG framework is described in Section 5.3, including code-mixed encoding, graph reasoning, quantum-enhanced retrieval, and explanation generation. Ablation results in Section 5.6 quantify each component's contribution. Anonymous source code (to be provided in supplementary material) includes model training scripts, evaluation pipelines, and the Gradio-based UI demo.

- **Datasets:** The HiFACT dataset of approximately 1,500 claims is described in Section 3, including collection, annotation, and distribution details. Class imbalance and challenges are explicitly discussed in Section 1. Dataset splits (train/validation/test) are detailed in Section 5.2 with stratification for fairness.

- **Experimental Setup:** Training, validation, and testing framework is documented in Section 5.1. Baselines (mBERT, IndicBERT, CM-BERT, VerT5erini, GPT-4, LLaMA-2, Mistral) are described in Section **??**. Evaluation metrics (Accuracy, Macro-F1, ROUGE-L, BLEU) are specified in Section 5.4. Human evaluation procedure (150 explanations, 3 annotators) is also provided in Section 5.4.

- **Theoretical Results:** Graph-based reasoning and quantum retrieval components reference prior theoretical work (e.g., Schuld et al., 2015; Kerenidis & Prakash, 2019). Assumptions and limitations are discussed in Section 6.

- **Supplementary / Appendix:** Annotation guidelines, hyperparameters, and error analyses are included in the Appendix (Section **??**). Screenshots of the Gradio UI (Figure 3) and pipeline diagram (Figure 2) aid interpretability.

## REPRODUCIBILITY STATEMENT

We have taken multiple steps to ensure the reproducibility of HiFACTMix. The dataset, methodology, and evaluation framework are comprehensively documented: the HiFACT dataset and its annotation pipeline are described in Section 3, while the architecture of HiFACTMix-Quantum-RAG, including graph reasoning and quantum-enhanced retrieval modules, is detailed in Section 5.3. The experimental setup, dataset splits, baselines, and evaluation metrics are provided in Section **??**, with ablation studies highlighting individual component contributions in Section 5.6. Hyperparameters, annotation guidelines, and additional error analyses are included in the Appendix (Section **??**). To support replication, we will release anonymized source code and dataset scripts as supplementary material. Together, these resources ensure that researchers can reproduce and extend our results reliably.

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
