# OpenReview forum: "HiFACTMix: A Code-Mixed Benchmark and Graph-Aware Model for EvidenceBased Political Claim Verification in Hinglish"
_ICLR.cc/2026/Conference — ICLR 2026 Conference Desk Rejected Submission_

### Official Review · Reviewer_LJ8D · 2025-10-30

**Soundness:** 1
**Presentation:** 1
**Contribution:** 1
**Rating:** 0
**Confidence:** 5

**Summary:**

This paper proposes a code-mixed Hinglish fact checking dataset of 1500 political claims with evidence documents and veracity labels, as well as a new RAG framework for the task that outperforms various BERT and T5 baselines.
Previous automatic fact checking efforts do not specifically model code mixing, which is the core novelty of this paper.

**Strengths:**

Previous automatic fact checking efforts do not specifically model code mixing, which is the core novelty of this paper.

**Weaknesses:**

- The dataset collection process is underspecified -- the paper states that "claims were collected from diverse sources", but the exact list of sources and how the claims were collected or selected is not specified. It is also not specified how the evidence documents are collected.
- Similarly, the annotation process is underspecified -- the veracity labels and meta-data categories are listed, and it is stated that annotations are performed by multiple reviewers, but nothing related to annotator selection, training, annotation quality etc. is described
- The method is only described in broad details -- Figure 2 shows the different components, and the various subsections of Section 4 provide references to methods which are re-used, but the description is not stand-alone, and crucial details such as how the models are trained are left out
- The paper claims that the RAG method is "quantum enhanced", but it is not explained what that means and why it is specifically relevant for code mixing
- There is an introduction of a simple user interface for a "demonstration" -- it is not clear what the relevance of this to the paper is. Is this being used to collect annotations? Have you evaluated it against other user interfaces?
- Some references under "recent LLMS" in Section 5.3 are not resolving
- The proposed model outperforms various BERT models and a T5 baseline, but not more recent models, though a core reason for that is likely because it is T5 based. Why did you not experiment with other LLMs are base models for your approach?

**Questions:**

See weaknesses

---

### Official Review · Reviewer_YQJo · 2025-10-30

**Soundness:** 1
**Presentation:** 1
**Contribution:** 1
**Rating:** 0
**Confidence:** 4

**Summary:**

This paper introduces HiFACTMix, a benchmark dataset of approximately 1,500 annotated Hinglish (Hindi-English code-mixed) political claims, each paired with evidence and veracity labels. The authors propose a fact-checking pipeline that integrates code-mixed text encoding, graph-based reasoning, a 'quantum-enhanced' RAG module, and explanation generation. The system is evaluated against multilingual and code-mixed baselines as well as recent LLMs, reporting improvements in explanation quality and accuracy. The paper also discusses ethical considerations and provides a user-facing demo.

**Strengths:**

- This paper studies a unique problem, that is, fact-checking in code-mixed (Hinglish) political discourse.
- This paper introduces a new annotated dataset (HiFACTMix) for Hinglish political claims, which could be useful.

**Weaknesses:**

- The presentation quality is poor. For example, the authors copied and pasted Figure 1 from (Guo et al., 2022). Please consider making original figures and using them in the paper.
- Key methodological details are missing or vague: the architecture, training pipeline, evidence graph construction, and integration of components are described only at a high level, with missing references and incomplete sections.
- Annotation protocol and inter-annotator agreement statistics are not described, raising concerns about dataset quality.
- Some references are speculative or incomplete, and several sections are unpolished (e.g., 'Section ??').

**Questions:**

- What are the annotation guidelines for the dataset, and what is the inter-annotator agreement for veracity and evidence alignment?
- Can the authors provide a more robust evaluation of explanation quality (e.g., using factual consistency metrics or detailed human evaluation protocols)?

**Details Of Ethics Concerns:**

Figure 1 is directly copied from an existing paper (Guo et al., 2022), which is subject to copyright compliance.

---

### Official Review · Reviewer_zeCe · 2025-10-31

**Soundness:** 2
**Presentation:** 1
**Contribution:** 2
**Rating:** 2
**Confidence:** 4

**Summary:**

This paper introduces HiFACT, a benchmark dataset of approximately 1,500 annotated Hinglish political claims from 28 Indian state Chief Ministers and other leaders (with veracity labels, evidence, and metadata), and proposes HiFACTMix, a graph-aware framework integrating quantum-enhanced RAG, code-mixed text encoding, and explanation generation.
Experimental results show HiFACTMix outperforms multilingual/code-mixed baselines (e.g., CM-BERT, VerT5erini) and is competitive with LLMs like GPT-4, with more evidence-grounded explanations.

**Strengths:**

- The paper fills the gap in fact-checking for code-mixed low-resource languages like Hinglish.
- The HiFACTMix framework not only outperforms strong multilingual and code-mixed baselines (such as CM-BERT and VerT5erini) in both veracity prediction and explanation quality but also remains competitive with advanced LLMs like GPT-4

**Weaknesses:**

At least 7 citation errors exist (e.g., Section 5.3, Reproducibility Checklist, Reproducibility Statement).

Insufficient dataset size (1.5k samples) and lack of clear description of annotation standards/quality.

No novelty in using widely-used RAG for empirical validation.

The HiFACTMix framework’s integration of graph-based reasoning and quantum-enhanced retrieval introduces substantial computational overhead that requires careful consideration.

Experiments are only conducted on self-constructed data without out-of-distribution evaluation, leading to untrustworthy results (potential overfitting).

Missing citations for baselines.

**Questions:**

see weaknesses.

---

### Official Review · Reviewer_4N1m · 2025-11-02

**Soundness:** 2
**Presentation:** 2
**Contribution:** 2
**Rating:** 2
**Confidence:** 4

**Summary:**

This paper introduces HiFACTMix, a Hinglish (Hindi–English) political fact-checking benchmark (~1,500 claims) with evidence and four-way veracity labels (True/False/Partially True/Unverifiable), and proposes a graph-aware, code-mixed fact-checking pipeline with a Quantum-Enhanced Retrieval-Augmented Generation (Quantum-RAG) module and an FLAN-T5–based explanation generator. The method combines a code-mixed encoder, an evidence graph with transformer-style reasoning, quantum-inspired re-ranking for retrieval, and natural-language justifications linked to retrieved evidence. Experiments compare against multilingual/code-mixed baselines (mBERT, IndicBERT, CM-BERT, VerT5erini) and recent LLMs (GPT-4, LLaMA-2, Mistral); the authors report stronger explanation quality and competitive overall results, plus ablations attributing gains to graph reasoning and Quantum-RAG.

**Strengths:**

1.	The primary strength of this paper is the introduction of the HiFACTMix dataset. Fact-checking in low-resource, code-mixed environments is a critical research gap, and this dataset, sourced from real-world political discourse, provides a much-needed resource for the community.
2.	The paper addresses a highly relevant and impactful problem. Hinglish is a dominant language in online political and social discourse in India, making it a key vector for misinformation. Developing tools for this specific context is a significant contribution.
3.	The proposed model attempts to build a complete fact-checking pipeline, from claim encoding and evidence retrieval to veracity prediction and justification generation. This holistic approach is commendable.
4.	The authors compare their model against a reasonable set of baselines, including those specific to code-mixed text (CM-BERT) and large-scale, general-purpose LLMs.

**Weaknesses:**

1.	The "Quantum-Enhanced RAG" is presented as a novel contribution but is poorly motivated and explained. The paper provides no specific details on the algorithm used, how it is "quantum-inspired," or why this approach is superior to established classical re-ranking methods (e.g., BM25, dense retrievers, or monoT5-based re-rankers). The ablation study merely shows that removing the component (i.e., using a simpler retrieval) hurts performance, but it provides no evidence that the "quantum" aspect itself adds any value over a standard, non-quantum-branded re-ranker.
2.	The paper relies on ROUGE and BLEU to evaluate explanation quality. The authors rightly acknowledge this as a limitation, but it is a significant one. These metrics measure surface-level n-gram overlap and are poor proxies for factual correctness, faithfulness to the evidence, or logical coherence. Given that RAG systems are known to be vulnerable to "fabricated content" and that their outputs must be evaluated for "Faithfulness", this evaluation is too shallow.
3.	The paper introduces a new static benchmark. However, a key challenge for RAG systems is robustness to "polluted" or adversarial knowledge bases. The evaluation does not test the model's resilience to "adversarial distractors"—documents that are semantically similar but factually incorrect.
4.	The dataset includes "Partially True" claims. This implies that claims may contain multiple facts that need to be verified independently. The paper's methodology (Figure 2) depicts a direct end-to-end model. It does not discuss or seem to implement a "Decompose-Then-Verify" strategy, which is often necessary for handling such complex, multi-fact claims. It is unclear how the model differentiates between "False" and "Partially True" without an explicit decomposition step, a process which itself is a known challenge due to potential error propagation.

**Questions:**

1. Can you provide a detailed technical explanation of the "Quantum-Enhanced RAG" module? What specific algorithm is being used, and how does it differ from a standard, SOTA classical re-ranker? Can you provide an ablation study comparing it to a strong classical re-ranker, not just a weaker baseline?

2. How does your model architecture handle "Partially True" claims? Does it perform any internal claim decomposition (as discussed in the "Decompose-Then-Verify" paradigm)? If not, how does it learn to distinguish a "Partially True" claim from a "False" one?

3. Given the known limitations of ROUGE/BLEU for this task, did you consider a human evaluation of explanation faithfulness (i.e., are the explanations factually grounded in the retrieved evidence)?

4.  Code-mixed handling. How do you tokenize romanized Hindi and mixed scripts? Any transliteration/normalization steps? What proportion of claims/evidence are fully romanized vs Devanagari/English?

5. Provide rater guidelines, agreement statistics, and a sample of evidence-linked explanations used in the study to verify faithfulness claims.

6. Which nodes/edges are extracted automatically, and with what tools (NER/linking/temporal parsing)?

7.  What is the size/domain/timeframe of the corpus?

---

### Note · Program_Chairs · 2026-01-17
**Submission Desk Rejected by Program Chairs**

The following references in this submission do not refer to real documents and/or have major errors in bibliographic information:

 Khyathi Chandu, Arijit Sharma, et al. Hiner: Hierarchical named entity recognition for code-mixed social media text. In Proceedings of EMNLP, 2018. P. Boonsanong, H. Kim, and X. Zhao. Fact-checking in the generative ai era: Challenges and opportunities. In Proceedings of ACL 2025, 2025.